# Delirium Screening and Pharmacotherapy in the ICU: The Patients Are Not the Only Ones Confused

**DOI:** 10.3390/jcm12175671

**Published:** 2023-08-31

**Authors:** F. Eduardo Martinez, Rebecca Tee, Amber-Louise Poulter, Leah Jordan, Liam Bell, Zsolt J. Balogh

**Affiliations:** 1Intensive Care Unit, John Hunter Hospital, Lookout Road, New Lambton Heights, Newcastle, NSW 2305, Australia; ed.martinez@health.nsw.gov.au (F.E.M.); rebecca.tee@health.nsw.gov.au (R.T.); amberlouise.poulter@health.nsw.gov.au (A.-L.P.); leah.jordan@hotmail.com (L.J.); liammichael.bell@health.nsw.gov.au (L.B.); 2School of Medicine and Public Health, University of Newcastle, Callaghan, NSW 2308, Australia; 3Department of Traumatology, John Hunter Hospital, Lookout Road, New Lambton Heights, Newcastle, NSW 2305, Australia; 4Injury and Trauma Research Program, Hunter Medical Research Institute, Newcastle, NSW 2305, Australia

**Keywords:** delirium, Intensive Care Unit, hospitalization, incidence, antipsychotic agents

## Abstract

*Background*: Delirium is difficult to measure in the Intensive Care Unit (ICU). It is possible that by considering the rate of screening, incidence, and rate of treatment with antipsychotic medications (APMs) for suspected delirium, a clearer picture can emerge. *Methods*: A retrospective, observational study was conducted at two ICUs in Australia, between April and June of 2020. All adult ICU patients were screened; those who spoke English and did not have previous neurocognitive pathology or intracranial pathology were included in the analysis. Data were collected from the hospitals’ electronic medical records. The primary outcome was incidence of delirium based on the use of the Confusion Assessment Method for ICU (CAM-ICU). Secondary outcomes included measures of screening for delirium, treatment of suspected delirium with APMs, and identifying clinical factors associated with both delirium and the use of APMs. *Results*: From 736 patients that were screened, 665 were included in the analysis. The incidence of delirium was 11.3% (75/665); on average, the Richmond Agitation and Sedation Scale (RASS) was performed every 2.9 h and CAM-ICU every 40 h. RASS was not performed in 8.4% (56/665) of patients and CAM-ICU was not performed in 40.6% (270/665) of patients. A total of 17% (113/665) of patients were prescribed an APM, with quetiapine being the most used. ICU length of stay (LOS), APACHE-III score, and the use of alpha-2 agonists were associated with the presence of delirium, while ICU LOS, the use of alpha-2 agonists, and the presence of delirium were associated with patients receiving APMs. *Conclusions*: The incidence of delirium was lower than previously reported, at 11.3%. The rate of screening for delirium was low, while the use of APMs for delirium was higher than the incidence of delirium. It is possible that the true incidence is higher than what was measured. Critical prospective assessment is required to optimize APM indications in the ICU.

## 1. Introduction

Delirium is a common neuropsychiatric syndrome characterized by acute fluctuations in mental status, inattention, and either disorganized thinking or an altered level of consciousness [1]. It is now widely accepted that the aetiology of delirium is likely multifactorial, with older age and cognitive impairment being some of the factors associated with delirium [2,3]. Other predictive factors include the presence of more than one condition associated with coma, sedative medications (like benzodiazepines), analgesics (like opioids and ketamine), increased severity of illness, type of admission (increased in emergency compared to planned), admission diagnosis (like neurological, neurosurgical, and trauma), the presence of infection, dehydration, malnutrition, renal failure, and the use of haemofiltration [4,5]. The pathophysiological pathway of delirium may depend on the cause. There are likely several pathways for the development of delirium, including GABAergic and cholinergic neurotransmitter systems [2]. This is suggested by the increased risk associated with GABA agonists and anticholinergic drugs [6]. Other pathways may include excess dopaminergic activity and inflammatory cytokines [2,4]. Delirium can be classified as hyperactive, hypoactive, or mixed, depending on the features of the symptoms [7]. The incidence of delirium in Intensive Care Units (ICUs) varies widely, ranging from 20% to 80%, depending on the patient population and the diagnostic criteria used [8,9]. Delirium in the ICU is associated with longer hospital stays, increased healthcare costs, and higher morbidity and mortality rates [10]. Prevention, early identification, and treatment of delirium are crucial when it comes to improving patient outcomes [10].

There are many tools used to screen for delirium, but The Richmond Agitation and Sedation Scale (RASS) and the Confusion Assessment Method for the Intensive Care Unit (CAM-ICU) are two of the most widely used [11,12]. Both screening tests are well-validated and are usually used together to assess sedation and diagnose delirium in ICU patients [13]. However, studies have shown that screening for delirium is inconsistent. This leads to delirium often going unrecognized and underdiagnosed in the ICU setting [14].

RASS and CAM-ICU are used in conjunction. RASS is used first to measure the level of arousal of patients. It can be used regardless of whether patients are receiving sedatives or not. The RASS score ranges from a score of −5, where a patient is deemed “unarousable”, to a score of +4, where a patient is “violent and immediate danger to staff”. If the RASS score is too low (less than −3), meaning that the patient is too drowsy or sedated, then the CAM-ICU cannot be performed [12]. The CAM-ICU test measures whether a patient is suffering from delirium at the time that the test is performed [12]. It uses a series of questions and commands to check if a patient is able to concentrate and undertake abstract thinking. Once delirium is identified, it is possible to grade its severity by using the CAM-S scale. By doing so, it is possible to determine its clinical trajectory as either improving or worsening [15].

Delirium is treated through both non-pharmacologic and pharmacologic strategies [16]. When pharmacologic strategies are needed, antipsychotic medications (APMs) are commonly prescribed, although their efficacy remains uncertain [17]. Some studies have suggested that APMs may be effective in reducing the duration and severity of delirium symptoms, while others have found no significant benefit [18,19]. The American Geriatrics Society guidelines for the management of delirium recommend the use of antipsychotic medications for patients with severe agitation or distress, but not for the routine treatment of delirium [20]. Despite the lack of definitive evidence and their considerable side effect profile, the use of APMs for delirium management in ICU patients is widespread [21].

By looking simultaneously at the patterns of screening for delirium, rates at which it occurs, and the treatment with APMs that patients with suspected delirium receive, clinicians might be able to obtain a more comprehensive picture of the overall landscape of delirium in the critical care setting. This more comprehensive evaluation might help determine whether the rates at which delirium is being identified are accurate and whether treatment is being administered appropriately. We hypothesize that screening rates for delirium are low and that the rates at which delirium is being identified are not similar to the rates at which delirium is being treated with APMs.

The overall aim of this study is to provide a comprehensive description of how critical care patients are burdened by delirium by looking at multiple parameters simultaneously, rather than a single measure. This study intends to achieve this aim by completing the objectives of describing the frequency of screening for delirium, describing the rates of delirium, including the incidence, and describing the treatment of suspected delirium with APMs. 

## 2. Materials and Methods

This was a retrospective, observational study, conducted at two metropolitan ICUs in Australia. Patients admitted to the two participating ICUs between 1 April 2020 and 30 June 2020 were reviewed. Inclusion criteria were adult patients of 18 years of age or older, admitted to the John Hunter Hospital ICU or Calvary Mater Newcastle ICU, who could be assessed for delirium at least twice daily with RASS and CAM-ICU tools. Exclusion criteria were age less than 18 years old; comatose, defined as a Glasgow Coma Scale (GCS) score of less than 8 for reasons other than purely sedation; non-English-speaking; patients with known pre-existing psychiatric disorders already receiving APMs; and patients with pre-existing cognitive impairment. These criteria made it possible to identify patients that could be assessed adequately with the tools being utilised and who would be at risk of suffering from delirium and not another type of neurological dysfunction from a primary pathology like dementia, schizophrenia, or a traumatic brain injury.

The primary outcome of the study was the incidence of delirium in ICU. This was considered as the number of patients with a positive CAM-ICU result out of those who were not previously known to have neurocognitive failure before presenting to ICU (i.e., new cases). The secondary outcomes were as follows: 1. the rates of use of RASS and CAM-ICU; 2. the rates of delirium based on a positive CAM-ICU result out of those who completed CAM-ICU assessments, therefore providing a period prevalence; 3. the rates of administration of APMs in the ICU for the treatment of suspected delirium; 4. the correlation of clinical parameters with the incidence of delirium; and 5. the relationship between clinical indicators and the propensity for the administration of APMs in suspected delirium cases.

The hospitals’ Electronic Medical Record (EMR) systems were used to screen and identify patients. All data were de-identified and collected using a password-protected electronic database. Only the study team had access to the data and database. An effort to minimize selection bias was made by screening all patients who were admitted to the participating ICUs. The sample size was a convenience sample, determined by the number of patients who were admitted during the pre-determined study period and that were eligible. The study protocol was approved by the local Human Research Ethics Committee and Governance offices before it was initiated (AU202009-13; AU202103-13). 

Patient data collected included demographics and baseline characteristics, treatment received in ICU, and outcome data. A measure of severity of illness was obtained by collecting Acute Physiology and Chronic Health Evaluation (APACHE) III scores. APACHE-III scores take into account a variety of clinical parameters and chronic health conditions within the first 24 h of admission to the ICU and are used to assess the prognosis of patients admitted to the ICU [22]. Sedation and delirium data were collected in the form of the RASS and CAM-ICU assessments performed by medical and nursing staff. Only data available for collection in the EMR were used. Since data regarding the type of delirium or severity of delirium were not routinely collected in the participating units, they did not form a part of this study.

Descriptive statistics were used with percentages and fractions. For determining which clinical factors are associated with delirium and with the use of APMs, multiple logistic regression was used. A *p*-value of <0.05 was considered significant.

## 3. Results

There were a total of 736 patients admitted to the participating ICUs during the study period. A total of 665 patients who met the inclusion criteria and had none of the exclusion criteria were included in the analysis (Figure 1). John Hunter Hospital is a tertiary-referral centre which provides comprehensive care with subspecialty teams that include cardiothoracic and vascular surgery, neurosurgery, interventional neuroradiology, trauma care, and extracorporeal therapies like haemodialysis, plasmapheresis, and ECMO. The Calvary Mater Newcastle Hospital is also a referral centre. It provides tertiary level care, including oncology, toxicology, and acute mental health illness services. The patients screened for this study were a cross-section of the population serviced by both hospitals’ ICUs.

The median age of the cohort was 64 (IQR 53–74) years, 62% (417/665) of the patients were male, the median APACHE-III score was 55 (IQR 40–75.5), and the median ICU LOS was 1.9 (IQR 1–3.6) days. The cohort’s baseline characteristics can be seen in Table 1.

The incidence of delirium, the primary outcome of this study, was 11.3% (75/665). Regarding screening for delirium, there were a total of 17,273 episodes of RASSs performed and a total of 1248 episodes of CAM-ICU assessments performed. This represents 7.2% (1248/17,273) of RASS assessments that go on to have a CAM-ICU assessment. RASS was performed an average of 8.2 times per patient per ICU day, while CAM-ICU was performed an average of 0.6 times per patient per ICU day. The primary outcome and secondary outcomes of screening can be seen in Table 2.

Regarding the treatment for suspected delirium, in this cohort, 17% (113/665) of patients received an APM for the treatment of suspected or diagnosed delirium. Of the patients that received APMs, 36.3% (41/113) had not been screened for delirium. Of the patients who were screened for delirium and received an APM, 9.1% (29/320) had a negative CAM-ICU result. Quetiapine was the APM most used as a first-line agent, which was administered to 6% (40/665) of patients, followed by haloperidol, which was administered to 4.5% (30/665) of patients. When a second- or third-line agent was needed, haloperidol was the APM most used. The maximum recommended dose was administered to only 1.2% (8/665) of patients [23]. The use of APMs for the treatment of suspected delirium in the ICU can be seen in Table 3.

Logistic regression was used to analyse the relationship between delirium and selected variables including age, gender, ICU LOS, admission type, APACHE-III score, and use of invasive mechanical ventilation (IMV), sedatives, benzodiazepines, alpha-2 agonists, and opioids. The results can be found in Table 4.

Logistic regression was also used to analyse the relationship between the use of APMs and important clinical characteristics. The results can be found in Table 5.

## 4. Discussion

This study provides a comprehensive view of the rates of screening and incidence of delirium, the patterns of use of APMs to treat suspected delirium, and factors associated with delirium. By including patients from two different institutions, a total of 2109 ICU days were analysed.

The patients included in this study had baseline characteristics that are similar to those encountered in most ICUs in Australia and New Zealand, as reported by the Australia and New Zealand Intensive Care Society (ANZICS) Centre for Outcome Resource Evaluation (CORE) [24]. This suggests that these results can be considered as generalizable.

In this study, the incidence of delirium was 11.3%, which is lower than the 20% to 80% that has been reported widely across the literature [25,26,27]. By taking into account the rate of screening and considering the number of positive CAM-ICUs over the number of CAM-ICUs performed (the rate of delirium), we see that the frequency with which delirium presents itself increases. The rate increases from an incidence of 11.3% to a rate of almost 15%. This highlights how delirium is difficult to measure, likely because of its fluctuating nature; it is possible that, to capture the true burden of delirium in critical care, Period-Prevalence might be a better measure than incidence [28,29,30].

Assessments with RASS were undertaken, on average, around every 3 h, when it is suggested that RASS provides the best utility when it is used every 1 to 2 h [22,23]. Assessments with CAM-ICU were performed, on average, every 40 h, when it is recommended that it is measured every 12 h [12]. This lower-than-recommended rate of screening is likely why the rates of delirium in this study were lower than those reported by other studies. 

CAM-ICU was performed on only 7.2% of the occasions that a RASS was performed. As mentioned previously, if the RASS score is less than −3, then CAM-ICU cannot be performed because the patient is too sedated. The RASS score was less than −3 only 6.2% of the time, which means that the low rate of CAM-ICU assessments was unlikely to be due to oversedation. It can be hypothesized that other factors were preventing its use. It is difficult to determine what factors specifically from these data alone; thus, further research is needed. 

When patients were suspected of or diagnosed with delirium and pharmacological treatment was utilised, an APM was prescribed 17% of the time. This rate is already higher than both the incidence and the rate of delirium found in this cohort. This makes us suspect, once more, that the burden of illness from delirium is higher than what is suggested by the previous figures. More than one third of the patients who received APMs had not been assessed for delirium with CAM-ICU. Along with that, in patients who had been screened for delirium but had a negative CAM-ICU result, almost 10% of these received an APM. It is difficult to know from these data alone why APMs were being used in these circumstances. It is possible that they were suffering from delirium and that this was recognised by clinicians, but this was not captured by the screening. This would also suggest that there is a significant unreported burden of illness from delirium in the critically ill population. 

In this cohort, APMs were used frequently and even on some patients who had no delirium. Along with APMs, it is important to note that patients suspected of delirium received non-APM treatments for delirium. Of the patients with a positive CAM-ICU result, 68% received benzodiazepines, most commonly as infusions. This was likely carried out for sedation while receiving IMV. Benzodiazepines have been associated with increased rates of delirium so it is surprising to see how their use is still relatively prevalent [27,31]. Of the patients with a positive CAM-ICU result, 45% received alpha-2 agonists as either boluses with clonidine or infusions with dexmedetomidine. Dexmedetomidine has shown promise for treating delirium, but the evidence is mixed [25,26,27]. It was unexpected that the rate of treatment with both APM and non-APM medications was higher than the actual documented rates of delirium. 

When attempting to identify factors that are associated with delirium, in this cohort only three were statistically significantly associated with delirium: ICU LOS, APACHE-III score, and the use of alpha-2 agonists. The factors analysed are all tightly interconnected, therefore making it difficult to determine how individual factors can increase the likelihood of developing delirium. Previous findings vary in terms of factors associated with delirium in the ICU [19,28]. With longer ICU LOS and a higher APACHE-III score, it is easy to see how organ failure like delirium can occur. It is possible that those with delirium were more likely to be prescribed alpha-2 agonists, rather than these causing the delirium. The timing of prescribing can help determine this in future studies. The only potentially modifiable factor associated with delirium, as long as the temporal relationship coincides, is the use of alpha-2 agonists. Future studies can continue to look for modifiable factors associated with delirium to help improve patient outcomes.

APMs are not without adverse effects; some of them, like prolonged QT intervals and dystonic reactions, are serious [29,30]. Minimizing their use puts patients at less risk, which is why identifying characteristics associated with a higher likelihood of receiving APMs could help clinicians plan how to best manage delirium when it occurs. From these data, we can see that a longer ICU LOS is associated with a higher chance of receiving APMs. Delirium itself is associated with a higher chance of receiving APMs, but receiving an alpha-2 agonist is also strongly associated. This means that, in the current environment, patients with delirium who have already received an alpha-2 agonist and continue to suffer from delirium are the most likely of all to receive an APM.

Some of the strengths of this study are that, by looking at delirium holistically (screening, rates, and treatment), we can potentially extract stronger insights about the complete landscape of this type of organ failure in the critical care population. This is a characteristic that makes this study unique when compared to most others, as it is common to focus on one aspect of the illness rather than looking at it from different angles. This can be more insightful for clinicians than a unidimensional perspective. Another strength is that including more than one centre makes the findings more generalisable. The limitations of this study are the same as those of other retrospective observational studies. While this is a limitation, the retrospective nature of the study allows for an assessment of the true adherence to delirium screening, which would have been harder to assess in a prospective study due to the study potentially altering usual practice. This study was undertaken at the beginning of the COVID-19 pandemic, so it might be representative of a period in history where the priorities of critical care practitioners were different to what they normally are, and treating patients with suspected delirium could have been different in terms of relying more on heavy sedation. We did not collect data around how many of the patients included here were admitted to the ICU due to COVID-19, so we cannot determine how this would have affected outcomes. At the height of the pandemic, with resources stretched, no patient was recruited into any type of study. This would have been valuable information but was not within the scope of the study. Data around the type of delirium were not available for collection. How the type of delirium affects outcomes could have made this report stronger. The timing and indications of medications and their relationship to the suspected or diagnosed delirium are difficult to determine due to the retrospective design. We were interested in short-term outcomes, although long-term outcomes using functional assessment scales would have also been insightful but were not available to us.

## 5. Conclusions

The incidence of delirium in this study was 11.3%. The screening rates with RASS and CAM-ICU were lower than recommended. The use of APMs is more frequent than the rate at which delirium is being diagnosed. A critical prospective assessment is required to optimize APM use in the ICU. Through ongoing attempts at determining its true burden of illness, delirium might become a type of organ failure that is less confusing for clinicians.

## Figures and Tables

**Figure 1 jcm-12-05671-f001:**
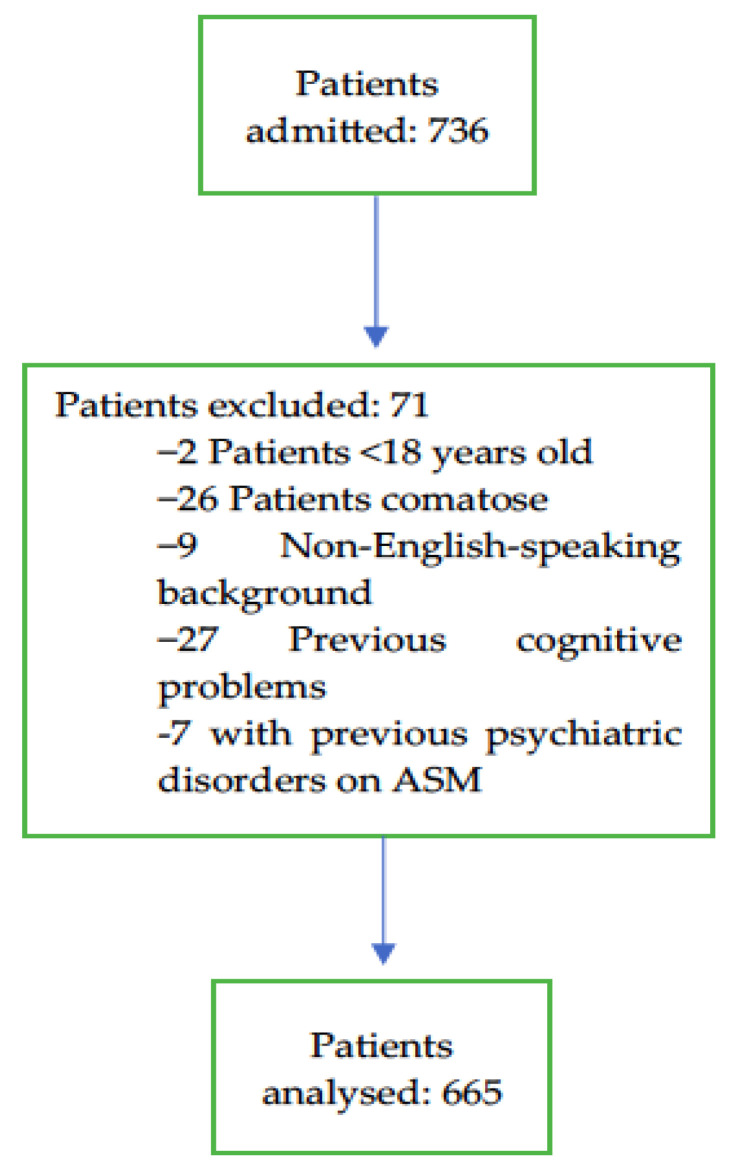
ICU admissions and exclusions, and the study population included in the final analysis.

**Table 1 jcm-12-05671-t001:** Patient baseline characteristics.

Characteristics	N = 665
Age	
Median (IQR)	64 (53–74)
Mean (SD)	61 (17)
Sex (male)	62.6% (417/665)
APACHE-III scores	
Median (IQR)	55 (40–75.5)
Mean (SD)	59 (26)
Admission type	
Medical	48.6% (323/665)
Surgical	51.4% (342/665)
ICU LOS (days)	
Median (IQR)	1.9 (1–3.6)
Mean (SD)	3.2 (5)
ICU LOS (hours)	
Median (IQR)	46 (24–86)
Mean (SD)	76.2 (120)
Received IMV at one point during ICU admission	
Yes	55.9% (372/665)
No	44.2% (293/665)
Sedative, any	57.1% (380/665)
Benzodiazepine	43% (286/665)
Alpha-2 agonist	16.7% (111/665)
Opioid	74% (492/665)
ICU mortality	10.5% (70/665)

APACHE-III: Acute Physiology and Chronic Health Evaluation- III; ICU: Intensive Care Unit; IMV: Invasive Mechanical Ventilation; IQR: Interquartile Range; LOS: length of stay; SD: Standard Deviation.

**Table 2 jcm-12-05671-t002:** Rates of delirium and frequencies of screening with RASS and CAM-ICU.

**Primary Outcome**	
Incidence, or rate, of patients who were found to have delirium at least once during their admission.	11.3% (75/665)
**Secondary Outcomes of screening**	
Total RASSs performed	17,273
Number of RASSs performed per patient per day (average number of RASSs/average ICU LOS)	8.2
Number of RASSs completed per patient during their ICU stays	
Median (IQR)	12 (5–27)
Mean (SD)	26 (52)
Time between RASS score evaluation (Average number of RASSs/24 h)	2.9 h
Number of patients who were not assessed with RASS at any time during their ICU stays	8.4% (56/665)
RASS score value	
Median (IQR)	−0.2 (−1.2 to 0.0)
Mean (SD)	−0.55 (1.5)
Number of patients with a mean RASS score < −3 (cannot be assessed with CAM-ICU)	6.2% (41/665)
Total CAM-ICUs performed	1248
Number of CAM-ICUs performed per patient per day (Average number of CAM-ICUs/average ICU LOS)	0.6
Number of CAM-ICUs completed per patient during their ICU stays	
Median (IQR)	1 (0–2)
Mean (SD)	1.9 (3.1)
Time between CAM-ICU evaluation (Average number of CAM-ICUs/24 h)	40 h
Number of patients who were	
assessed with CAM-ICU at least once while in the ICU	59.4%(395/665)
not assessed with CAM-ICU at any time while in the ICU	40.6% (270/665)
Rate of delirium based on assessments (Positive CAM-ICU results/Total CAM-ICU results)	14.8% (185/1248)
Rate of patients who had at least one positive CAM-ICU result out of those who were assessed with CAM-ICU	19% (75/395)

CAM-ICU: Confusion Assessment Method-ICU; ICU: Intensive Care Unit; IQR: Interquartile Range; LOS: length of stay; RASS: Richmond Agitation Sedation Scale; SD: Standard Deviation.

**Table 3 jcm-12-05671-t003:** Treatment with antipsychotic medications in the ICU for suspected delirium.

**n = 665**	
APM prescribed	17% (113/665)
Patients who received APMs	
After being screened for delirium	63.7% (72/113)
After not being screened for delirium	36.3% (41/113)
Patients who were screened for delirium who received APMs	
with a positive CAM-ICU result	57.3% (43/75)
with a negative CAM-ICU result	9.1% (29/320)
Patients that received	
one agent	12.6% (84/665)
two agents	3.5% (23/665)
three agents	0.9% (6/665)
Most used APM	
Quetiapine	6% (40/665)
Haloperidol	4.5% (30/665)
Olanzepine	3.3% (22/665)
Droperidol	2.6% (17/665)
Risperidone	0.5% (3/665)
Other	0.2% (1/665)
Patients who received non-APM treatment with a benzodiazepine	
with a positive CAM-ICU result	68% (51/75)
with a negative CAM-ICU result	35% (114/320)
Patients who received non-APM treatment with an alpha-2 agonist	
with a positive CAM-ICU result	45.3% (34/75)
with a negative CAM-ICU result	13.4% (43/320)

APMs: antipsychotic medications; CAM-ICU: Confusion Assessment Method for ICU; ICU: Intensive Care Unit.

**Table 4 jcm-12-05671-t004:** Multiple logistic regression for the association of clinical factors with delirium.

Variable	Odds Ratio	95% CI (Profile Likelihood)
Intercept (Delirium)	0.012	0.002217 to 0.05621
Gender	1.119	0.5980 to 2.132
Age	1.003	0.9832 to 1.024
ICU LOS in hours	1.006	1.003 to 1.009
Admission type (surgical, medical)	0.8267	0.4034 to 1.683
APACHE-III score	1.027	1.013 to 1.041
Intubated	0.9344	0.4105 to 2.157
New sedative prescribed	1.385	0.4597 to 4.087
New benzo prescribed	2.007	0.8929 to 4.778
Alpha-2 agonist prescribed	2.760	1.302 to 5.919
New opioid prescribed	0.6006	0.2525 to 1.418

APACHE-III: Acute Physiology and Chronic Health Evaluation III; APM: antipsychotic medication; CI: Confidence Interval; ICU: Intensive Care Unit; LOS: length of stay.

**Table 5 jcm-12-05671-t005:** Multiple logistic regression for the association between clinical factors and the use of APMs.

Variable	Odds Ratio	95% CI (Profile Likelihood)
Intercept (APM use)	0.08668	0.02639 to 0.2658
Gender	1.055	0.6059 to 1.865
Age	0.9856	0.9696 to 1.002
ICU LOS in hours	1.003	1.001 to 1.006
Admission type (surgical, medical)	1.503	0.8428 to 2.712
APACHE-III score	1.002	0.9906 to 1.014
Intubated	0.6482	0.3438 to 1.196
Alpha-2 agonist (clonidine/dexmed) prescribed	7.615	4.182 to 14.04
Delirium	3.537	1.747 to 7.066

APACHE-III: Acute Physiology and Chronic Health Evaluation III; APM: antipsychotic medication; CI: Confidence Interval; ICU: Intensive Care Unit; LOS: length of stay.

## Data Availability

Data collected and presented here can be made available on direct request to the corresponding author.

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
