# Peer review of "Delirium Screening and Pharmacotherapy in the ICU: The Patients Are Not the Only Ones Confused"

_jcm, 2023, doi:10.3390/jcm12175671_

Round 1

Reviewer 1 Report

The study by F. Eduardo Martinez et al. is a retrospective study evaluating delirium and its characteristics in the ICU context. Although potentially interesting given the high sample size, it does not add much new information to what is already available in the literature on the subject. In addition, I have some suggestions for the Authors particularly on the methodological section: 

- Antipsychotic Medications (APM) is often replaced in the text with ASM (?), e.g. in lines 23, 31 and 33. I suggest maintaining consistency in the terminology used to avoid ambiguity;

- The introduction is very succinct, particularly in relation to the pathophysiological mechanisms related to delirium. Since various pharmacological agents are then considered in relation to delirium and its treatment, it would be useful to include some explanation of the molecular mechanisms involved in its pathogenesis and treatment, so as to justify the results obtained;

- The methodological part requires, in my opinion, major revisions: the RASS and CAM-ICU questionnaires should be explained more clearly and comprehensively. Moreover, the type of delirium examined (i.e. hyperactive, hypoactive or mixed) is never mentioned in the text. Which type of delirium is most frequent? Which one has a better outcome? Which one responds best to treatment (and if so, to which?)? The Authors also have as a secondary outcome the evaluation of clinical aspects, but no information is given about comorbidities (not even those that might have a pathophysiological connection or increase the risk of delirium) nor about the causes of admission itself. Is it possible to assess the most frequent causes of hospitalisation and their relation to the outcomes measured? Are there comorbidities that are more frequently associated with delirium and influence its outcome? 

- In Table 2 and the Results, the APACHE-III score is mentioned, which, however, is not introduced in the Methods. I suggest explaining it and evaluating its implications after describing it. 

- The Authors write that some patients received APM without having a diagnostic suspicion for delirium. In those cases, for what reason were these drugs administered? 

- In the Limits it is correctly assessed that the study was conducted in the context of the Covid-19 era (with all the associated changes in clinical practice that characterised this period) and various studies have investigated its relationship to the risk of delirium. How many of the patients investigated had contracted Covid-19 infection (prior or at the time of admission)? Was it associated with significant differences in the measures collected (e.g. intubation)? How did it change the outcome? Comparing it with other similar studies in the previous period, are there differences in the measures collected? 

- Is it possible to obtain information, e.g. with some functional assessment scales (e.g. Barthel Index) on the distant outcome of these patients (taking into account in particular the questionnaires and therapies used)? 

- Have any of the patients recruited performed typical ICU environment analyses to assess prognosis (e.g. N20, EEG)? Are there relationships with the investigated measures? 

Author Response

Thank you to our reviewer for the valuable feedback given to us. We have worked diligently to address each point. The reviewer's comments are in italics and our replies are in black:

The study by F. Eduardo Martinez et al. is a retrospective study evaluating delirium and its characteristics in the ICU context. Although potentially interesting given the high sample size, it does not add much new information to what is already available in the literature on the subject. In addition, I have some suggestions for the Authors particularly on the methodological section: 

- Antipsychotic Medications (APM) is often replaced in the text with ASM (?), e.g. in lines 23, 31 and 33. I suggest maintaining consistency in the terminology used to avoid ambiguity;

This has been corrected throughout the manuscript and only the abbreviation APM is now used consistently.

- The introduction is very succinct, particularly in relation to the pathophysiological mechanisms related to delirium. Since various pharmacological agents are then considered in relation to delirium and its treatment, it would be useful to include some explanation of the molecular mechanisms involved in its pathogenesis and treatment, so as to justify the results obtained;

Although the mechanistic aspects of delirium are far beyond the focus of this paper, the introduction has been expanded and now included more detail factors dealing to delirium and the pathophysiology for delirium.

- The methodological part requires, in my opinion, major revisions: the RASS and CAM-ICU questionnaires should be explained more clearly and comprehensively. Moreover, the type of delirium examined (i.e. hyperactive, hypoactive or mixed) is never mentioned in the text. Which type of delirium is most frequent? Which one has a better outcome? Which one responds best to treatment (and if so, to which?)? The Authors also have as a secondary outcome the evaluation of clinical aspects, but no information is given about comorbidities (not even those that might have a pathophysiological connection or increase the risk of delirium) nor about the causes of admission itself. Is it possible to assess the most frequent causes of hospitalisation and their relation to the outcomes measured? Are there comorbidities that are more frequently associated with delirium and influence its outcome? 

RASS and CAM-ICU are explained in more detailed in the Introduction section, therefore setting the scene for the Methods section. The types of delirium and measures for severity of delirium are now discussed. Because size and design of our paper would not allow to make conclusions about which pathology leads to delirium, we did not aim our study for this outcome. The cause for admission was not collected and only the type of admission (medical/surgical) information was available. This shortcoming is now listed in the limitation section of the manuscript.

- In Table 2 and the Results, the APACHE-III score is mentioned, which, however, is not introduced in the Methods. I suggest explaining it and evaluating its implications after describing it. 

APACHE-III has now been addressed in the Methods section to set the scene for the results found in Table 2.

- The Authors write that some patients received APM without having a diagnostic suspicion for delirium. In those cases, for what reason were these drugs administered? 

This is a very insightful comment by the reviewer, unfortunately this cannot be responsibly commented from this study design, but certainly generated a new hypothesis to explore. We have made some attempt to address your comment in the manuscript and is now included in Line 284.

- In the Limits it is correctly assessed that the study was conducted in the context of the Covid-19 era (with all the associated changes in clinical practice that characterised this period) and various studies have investigated its relationship to the risk of delirium. How many of the patients investigated had contracted Covid-19 infection (prior or at the time of admission)? Was it associated with significant differences in the measures collected (e.g. intubation)? How did it change the outcome? Comparing it with other similar studies in the previous period, are there differences in the measures collected? 

Again, very insightful by the reviewer, but this data was not collected as it was not within the scope of the study. It has been discussed in Line 343.

- Is it possible to obtain information, e.g. with some functional assessment scales (e.g. Barthel Index) on the distant outcome of these patients (taking into account in particular the questionnaires and therapies used)? 

This data is not available at our institutions and therefore not collected, although it would have been valuable to obtain and analyse. This has been addressed in Line 348.

- Have any of the patients recruited performed typical ICU environment analyses to assess prognosis (e.g. N20, EEG)? Are there relationships with the investigated measures? 

These tests were not performed within this cohort and are only rarely ordered at our institutions, therefore were not included in the analysis.

Reviewer 2 Report

Dear Authors,

I read your work entitled “Delirium screening and pharmacotherapy in ICU: the patients are not the only ones confused.” and here I enclose my recommendations to you:

1.     There is a need for editing some of English language errors. Please, have a more thorough “look” in the text.

2.     The “Introduction” section even has some information it has a shortage in references. Even it is clearly introducing the topic and the unique characteristics of this condition. however, it would benefit from a clearer statement of the objectives or the main points the authors intend to address. This would provide readers with a roadmap of what to expect throughout the manuscript. Generally, by elaborating the above sure it would provide a clearer rational of this study. I strongly suggest the Authors to address such issues.

3.     The “Methods” section is written very well but sure extra information’s are needed. I suggest the Authors to add more info about the participants. Furthermore, I suggest the Authors to remove from the “Results” section lines 105-134 and to add them to the “Methods” section. This will retain the “Results” section sound and clear.

4.     The “Results” are readers friendly and sure there is no need for new analysis.

5.     The “Discussion” section is well organised and written with adequate literature support. I congratulate the Authors for that.

6.     Please include limitations to this study.

Thank you.

 Minor editing of English language required

Author Response

Thank you to our reviewer for the valuable feedback given to us. We have worked diligently to address each point. The reviewer's comments are in italics and our replies are in black:

  1. There is a need for editing some of English language errors. Please, have a more thorough “look” in the text.

Thank you for your comment. We have done this and corrected some of the errors that were found throughout the manuscript. Another native English speaker reviewed the manuscript. Any further specific grammatical errors identified, please specify, we are happy to address.

  1. The “Introduction” section even has some information it has a shortage in references. Even it is clearly introducing the topic and the unique characteristics of this condition. however, it would benefit from a clearer statement of the objectives or the main points the authors intend to address. This would provide readers with a roadmap of what to expect throughout the manuscript. Generally, by elaborating the above sure it would provide a clearer rational of this study. I strongly suggest the Authors to address such issues.

Thank you again. We have attempted to address this issue by explaining to the reader what the overall aim of the study is and how we will attempt to achieve that aim through specific objectives. This can now be found towards the end of the Introduction in Line 92.

  1. The “Methods” section is written very well but sure extra information’s are needed. I suggest the Authors to add more info about the participants. Furthermore, I suggest the Authors to remove from the “Results” section lines 105-134 and to add them to the “Methods” section. This will retain the “Results” section sound and clear.

Thank for this comment. A description of the population that this cohort represents has been given in the text of the Results section. The diagnosis or subspecialty category of the patients included in the text was not collected, therefore cannot be provided in the Tables.

  1. The “Results” are readers friendly and sure there is no need for new analysis.

Noted with thanks.

  1. The “Discussion” section is well organized and written with adequate literature support. I congratulate the Authors for that.

Noted with thanks.

  1. Please include limitations to this study.

Limitations of this study are described in Line 357.

Reviewer 3 Report

I have thoroughly examined the manuscript entitled “Delirium screening and pharmacotherapy in ICU: the patients 2 are not the only ones confused.”. The manuscript is well-written. My specific comments are as follows:

Introduction

Introduction section is not fully establishing the case for this review. Like hypothesis in this section not very clear.

There should be some articles showing the in-depth analysis -at molecular level should be included to allure attention of wide gamut of readers.

Richmond Ag-44 itation and Sedation Scale (RASS) and the Confusion Assessment Method are considered as gold standard to diagnose the delirium but there must some drugs used in ICU that could affect the situation like delirium, authors should also mention something about it- may in table form with statistical analysis.

Material and Methods

Rationale of inclusion and exclusion criterion of patients can be elaborated.

Discussion

Authors can write 2-3 sentences about the uniqueness of this review over other.

Author Response

Thank you to our reviewer for the valuable feedback given to us. We have worked diligently to address each point. The reviewer's comments are in italics and our replies are in black:

Introduction

Introduction section is not fully establishing the case for this review. Like hypothesis in this section not very clear.

This has been addressed in line 91. A better description of the aims and objectives is also provided.

There should be some articles showing the in-depth analysis -at molecular level should be included to allure attention of wide gamut of readers.

There is now a description of the pathophysiology of delirium in the Introduction, which describes a hypothesis of the illness at a molecular level.

Richmond Ag-44 itation and Sedation Scale (RASS) and the Confusion Assessment Method are considered as gold standard to diagnose the delirium but there must some drugs used in ICU that could affect the situation like delirium, authors should also mention something about it- may in table form with statistical analysis.

This has been expanded on the Introduction section in Line 42. Including it in the analysis would have been beneficial for making the manuscript stronger, but this data was unfortunately not collected. The high risk ICU medications for delirium development are already widely published, we have referenced some now in Discussion.

Material and Methods

Rationale of inclusion and exclusion criterion of patients can be elaborated.

This has been included in Line 152.

Discussion

Authors can write 2-3 sentences about the uniqueness of this review over other.

This has been addressed in Line 362.

Round 2

Reviewer 1 Report

I thank the Authors for their extensive and precise review. I have only one last suggestion: I recommend the Authors to include in the Results (also in a table) the percentage of patients with the different types of delirium (i.e. hyperactive, hypoactive, mixed and indeterminate) and to evaluate in the discussion whether the outcomes assessed are correlated with one subtype in particular with respect to the others (e.g. are the hyperactive ones treated more than the hypoactive or mixed ones?). 

Author Response

We thank the reviewer for their valuable input. Suggestions are in italics and our response in black.

I thank the Authors for their extensive and precise review. I have only one last suggestion: I recommend the Authors to include in the Results (also in a table) the percentage of patients with the different types of delirium (i.e. hyperactive, hypoactive, mixed and indeterminate) and to evaluate in the discussion whether the outcomes assessed are correlated with one subtype in particular with respect to the others (e.g. are the hyperactive ones treated more than the hypoactive or mixed ones?). 

Thank you to the reviewer for this suggestion, which would definitely help improve our manuscript. Unfortunately, this data was not available for collection since the participating hospitals do not routinely collect this data. This is mentioned in the Methods section in Line 194.

We have also included a further point of discussion of how based on the rates of type of delirium described in the literature we expect that the same rates would occur here. Unfortunately, we cannot provide this information in a table, as we do not have it. We have included this in the limitations of the study.